# The Use of Mobile Sensors by Children: A Review of Two Decades of Environmental Education Projects

**DOI:** 10.3390/s23187677

**Published:** 2023-09-05

**Authors:** Maria João Silva, Cristina Gouveia, Cristina Azevedo Gomes

**Affiliations:** 1CIED, School of Education, Polytechnic Institute of Lisbon, 1549-003 Lisboa, Portugal; 2Liga para a Protecção da Natureza, 1500-124 Lisboa, Portugal; cristina.gouveia@lpn.pt; 3CI&DEI, School of Education, Polytechnic Institute of Viseu, 3504-501 Viseu, Portugal; mcagomes@esev.ipv.pt

**Keywords:** sensors, environmental education, mobile devices, multisensory, cross-case analysis

## Abstract

Over the past twenty years, the use of electronic mobile sensors by children and youngsters has played a significant role in environmental education projects in Portugal. This paper describes a research synthesis of a set of case studies (environmental education projects) on the use of sensors as epistemic mediators, evidencing the technological, environmental, social, and didactical dimensions of environmental education projects over the last two decades in Portugal. The triggers of the identified changes include: (i) the evolution of sensors, information and communication platforms, and mobile devices; (ii) the increasing relevance of environmental citizenship and participation; (iii) the recognition of the role of multisensory situated information and quantitative information in environmental citizenship; (iv) the cause–effect relation between didactical strategies and environmental education goals; (v) the potential of sensory and epistemic learners’ practices in the environment to produce learning outcomes and new knowledge. To support the use of senses and sensors in environmental education projects, the SEAM model was created based on the developed research synthesis.

## 1. Introduction

The final goal of the research presented in this paper is to create a visual model of the use of sensors in environmental education projects. An analysis of a set of environmental education projects implemented in Portugal is developed for that purpose.

This introduction outlines the main topics of the present research, connecting environmental education, human senses, and mobile electronic sensors.

The environmental education projects discussed in this study align with the overarching objective outlined in the Belgrade Charter: “To develop a world population that is aware of, and concerned about, the environment and its associated problems, and which has the knowledge, skills, attitudes, motivations and commitment to work individually and collectively toward solutions of current problems and prevention of new ones [1]” (p. 3). This objective underscores the action and agency of each person within the framework of environmental education.

The conceptual framework for integrating mobile electronic sensors into environmental education, as outlined in this paper, highlights the potential of these sensors, along with other digital technologies, to cultivate a deeper and more empathetic connection with both the environment and fellow individuals [2]. Human senses can be compared with biological sensors as they possess the capability to perceive physical, chemical, and biological characteristics [3]. Mobile electronic sensors can be used by young learners as extensions of their innate senses. Consequently, the collaborative use of human senses and electronic sensors emerges as a relevant strategy for educating young learners in and about the environment, echoing Arthur Lucas’s ideas in 1972 [4]. In his doctoral dissertation, Lucas distinguishes between environmental education “about” the environment, “in” the environment, and “for” the environment. The current research study primarily centers around employing sensors for the first two approaches rather than being focused on knowledge acquisition “about” the environment [4].

The joint use of senses and mobile electronic sensors by children and youngsters in the environment involves embodied experiences and facilitates the acquisition of information through perception as well as the quantitative data gathered by the sensors. Abstract knowledge is then synthesized and processed in the mind, rooted in the *praxis* of the bodies [5]. This localized, embodied exploration and reflection, extended by mobile electronic sensors, enables responsible and critical human agency and action within the sphere of environmental education [5].

Collecting and analyzing local environmental data plays a pivotal role in motivating children and youngsters to participate actively within their local communities [6]. To maximize the synergies between human senses and sensors, learners should exercise control over the sensors, encompassing factors such as timing, location, data acquisition mode, and visualization. As such, these sensors need to be tangible, robust, mobile, configurable, and easy-to-use, with user interfaces that are as transparent as possible. Such affordances are nowadays made possible by the availability of affordable kits of didactics sensors, which work together with apps.

When young individuals assume control over sensors and sensory input during environmental exploration, a sense of competence flourishes [7]. Furthermore, interactions with the environment foster identity and responsibility, enabling data monitoring, problem-solving, and decision-making [7].

The tangible information acquired via human senses and the abstract knowledge acquired by sensors are inherently tied to geographic locations. Consequently, the environmental education projects explored in this research study generate Voluntary Geographic Information (VGI), encompassing spatial data. The aim of this study was to encourage children and youngsters to voluntarily produce spatial data and contribute to scientific environmental knowledge [8].

Aligned with the overarching goal of environmental education, the incorporation of mobile electronic sensors into the learning process, in and for the environment, can be mediated to develop awareness, skills, and attitudes to address and prevent environmental problems.

The integration of electronic sensors into environmental education has primarily been driven by market forces that have made these tools accessible for everyday educational activities. Electronic sensors have evolved to become smarter, smaller, portable, affordable, and wireless [9,10].

Over the past two decades, sensor technologies have been moving towards specialization [11]. This trend is mirrored in the educational landscape, where a diverse array of sensor providers now offers products tailored for teaching and learning. Sensors designed for children encompass a wide range of environmental parameters, including air quality, water quality, soil quality, light intensity, sound levels, and more. These sensors empower students to collect real-time data and gain insights into various environmental aspects. These advancements offer educators and students enhanced opportunities to explore and comprehend the environmental factors in their surroundings.

Another significant trend impacting environmental education is the substantial reduction in sensor costs. This shift is particularly evident in air quality monitoring, where low-cost sensors worn or carried by individuals estimate one’s personal exposure to different types of pollution [12]. An example of such sensors being applied in citizen science projects and educational initiatives is the Geoair2.0 sensor [13], which is capable of measuring air quality variables such as particulate matter (PM) and volatile organic compounds (VOCs), as well as humidity and temperature. This kit also includes LTE-M, GPS, Wi-Fi, a long-lasting battery, and a display screen.

The proliferation of low-cost sensors has coincided with a marked increase in the use of Information and Communication Technologies (ICT) devices such as laptops, tablets, and smartphones in educational projects The latter have had a profound impact not only because they are considered an extension of the self by children and youngsters [14] but also due to the integrated set of sensors they possess. With the widespread use of smartphones, an array of sensors with educational potential have become accessible [15,16]. Notably, two sensor groups stand out: (1) cameras and microphones and (2) GPS. Cameras and microphones enable objective recordings of sensory data, while concurrent GPS data can be used to register and estimate the spatial distributions of environmental variables. Smartphones, in addition to their integrated sensors, function as interfaces for the Internet of Things.

The rise of the Internet of Things has introduced new possibilities, particularly with platforms like Arduino and Raspberry Pi [17,18,19]. These open-source platforms, closely aligned with the DIY movement, have fostered the development of educational projects within the realm of environmental education. Notably, these advances have led to the creation of kits that are specifically designed for education, emphasizing user-friendliness for both students and teachers. One example of such a kit is the Explore IoT Kit Rev2, an Arduino-based kit linked to the UN’s Sustainable Development Goals (SDGs). These kits include components such as data collection interfaces with an array of sensors and data analysis and processing tools while providing data visualization and communication capabilities [20]. By diminishing barriers to technology adoption, students and teachers can focus on exploring their environmental surroundings.

Beyond this introductory context, the subsequent section examines how the incorporation of sensors into environmental education projects can be framed across multiple dimensions. The third section of this paper synthesizes the research we conducted, and this is followed by the presentation and discussion of our results in Section 4 and Section 5, respectively. Section 6 presents the SEAM model, a direct outcome of the conducted research, and this is followed by Section 7, which contains concluding remarks. Finally, the last section provides a list of references.

## 2. The Multidimensional Use of Sensors by Children and Youngsters in Environmental Education Projects

With teacher mediation, young learners use mobile sensors in diverse ways and for different objectives within environmental education projects. The multiple dimensions of these projects, including technological, social, didactic, and epistemic aspects, influence how sensors are integrated into the activities of children and youngsters. This influence is described in the present section.

Regarding the technological dimension, as elucidated in the Introduction, electronic sensors have evolved to become became simpler, smaller, cheaper, mobile, and wireless [9,10]. They have become ubiquitous and usable everywhere, anytime [21]. During the early 21st century, smartphones introduced the capability to sense and record environmental and multisensory data in a familiar manner. This was facilitated by interfaces incorporating familiar metaphors [22] and by the incorporation of sensors such as GPS, microphones, cameras, and accelerometers [23]. More recently, tablets have gained popularity, particularly in educational settings, often supplanting mobile phones due to their larger screens, which enhance collaborative work, especially among younger users.

In 2019, the EU Kids Online network conducted a survey involving children aged 9–17 from 19 European countries, including 1974 children in Portugal [24]. Smartphones have emerged as the preferred means of accessing the internet (87%), surpassing computers by over twice the rate (41%). Their usage extends to 57% of 9–10-year-olds, 83% of 11–12-year-olds, and 95% of older children who own smartphones. Tablet usage was reported by 25% of the surveyed children, with a decline in usage observed with age.

In a recent study conducted with younger children aged 3–8 years [25], researchers conducted face-to-face interviews in 656 Portuguese households. The study involved a questionnaire for the parents of children aged between 3 and 8 years and a separate questionnaire for children aged 6–8 years. Their findings indicated that smartphones and televisions are the most prevalent screens in households, followed by tablets and laptops. All children engage with television, while half of them play digital games, and 38% use the internet. Internet access increases significantly with age, with 22% of 3–5-year-olds and 62% of 6–8-year-olds accessing the internet. Mobile devices—tablets, laptops, and smartphones—are the primary means of accessing the internet, with approximately half of the internet users having their own tablet.

In the early years of the second decade of the 21st century, the advent of mobile electronic sensors combined with data loggers ushered in a new era for environmental education. This innovation facilitated data collection, storage, and visualization in multiple representations [26]. These systems, following pioneering projects like MobGeoSens [27], have been integrated with smartphones and tablets in environmental education projects. Nevertheless, it is important to note that high-precision environmental sensors, such as those focused on air quality, are not yet standard features in everyday mobile devices.

Within the social dimension of environmental education projects, the significance of global environmental participation has grown, particularly since the emergence of Agenda 21 and the Rio Declaration on Environment and Development [28], which popularized the concept of “education for sustainable development”. In the Portuguese educational system, citizenship education is now intertwined with education for sustainable development, emphasizing participation and the interconnection of social, economic, and environmental aspects. As a result, project-based and problem-based learning have become increasingly crucial strategies in environmental education within schools [29].

Participation stands as the highest level among the environmental education objectives, alongside categories like awareness, knowledge, and skills, as originally outlined during the Tbilisi Conference [1]. Employing sensors within project- and problem-based learning demands intentionality and purpose, directly linked to problem-based inquiries and the required data collection necessary to address those inquiries.

From a didactic perspective, the development of a specific category of educational goals requires the use of targeted educational strategies. Experiential learning offers an embodied, enactive approach to fostering environmental awareness by facilitating a dialogue between experiential encounters and theoretical learning [30]. According to Haskell, within experiential learning, our experience stems from the collective synergies between embodied perceptions, environment, and social history [30]. This process engenders both a heightened empathy and a refined sensitivity towards the environment, alongside the assimilation of diverse forms of knowledge, which is achieved through sensory perception and experiential engagement, interaction with abstract concepts, reflective observation, and proactive exploration. When implementing the experiential learning strategy, the use of sensors by learners centers around the thorough exploration of their environment and its attributes, often providing quantitative augmentation to the outcomes of learners’ embodied and contextually grounded multisensory observations.

Research-based learning represents a student-centered model that promotes awareness, knowledge acquisition, and competencies. It integrates research methodologies into the learning process, treating research as a means to uncover, develop, test, and communicate knowledge [31]. While rooted in academic settings, this model has been implemented in primary and secondary schools [32], with increased support to foster autonomy.

Research-based learning combines critical thinking and inquiry, empowering students to take charge of their learning. In this model, inquiry can extend beyond bibliographic research to encompass problem-solving and experimental activities, often converging with problem-based and project-based learning. Within inquiries, sensor use centers around the stages of inquiry. Even bibliographic inquiries can be enriched through sensor integration. For instance, data collected by microphones and cameras can be used for bibliographic searches based on produced sounds and images.

Problem-based learning not only facilitates the acquisition of diverse forms of knowledge but also cultivates global competencies such as information searching and organization, the practical application of knowledge in everyday problems, the effective communication of processes and results, the selection of appropriate problem-solving techniques, and collaborative and autonomous work [33]. Problem-based learning is a collaborative process that encompasses formulating relevant problem statements; identifying data sources; collecting, processing, and interpreting data; reaching problem resolutions; and assessing outcomes while also identifying analogous problems and scenarios [34].

In situated problem-based learning, sensor utilization assumes a central role within environmental education projects, as sensors enable the detection and, often, the measurement of environmental variables. This allows for data collection, which is necessary to address problem statements and arrive at evidence-based conclusions. Sensors also facilitate the creation of multiple data visualizations, aiding in communicating processes and outcomes.

Project-based learning can be developed together with problem-based learning. The distinctive factor of project-based learning lies in the requirement for real-world, situated problem statements and the extended timeframe for implementation [35].

Epistemically, the application of diverse didactic strategies engenders the development of distinct epistemic practices among learners—practices that generate knowledge [36]. These practices encompass observation, data collection, recording, interpretation, data-driven conclusions, and the communication of processes and outcomes. The use of sensors can scaffold these epistemic practices, serving as epistemic mediators. With teacher mediation, sensors facilitate the understanding of qualitative and quantitative environmental variables, bridging the visible and invisible realms and connecting concrete experiences with abstract concepts [37].

Across all aforementioned didactic strategies, learners can leverage everyday and laboratory resources, including sensors, as epistemic mediators—external tools utilized to construct knowledge [38]. Learners interact with these resources, linking them to everyday experiences and concepts, thereby facilitating knowledge acquisition and creation [39].

## 3. Research Design

The central goal of this research study was to develop a visual model of the use of sensors within environmental education projects. Our specific research goals included the following: (i) to outline the underlying dimensions of environmental education projects, (ii) to analyze the influence of the projects’ dimensions in the use of sensors, (iii) to create categories of thematic analysis based on the previously defined dimensions, (iv) to classify both common and distinctive features/themes in the use of sensors across environmental education projects using the defined categories of analysis, and (v) to create and organize clusters of fundamental features/themes in a visual model.

To achieve these goals, we conducted an in-depth analysis of sensor use in the environmental education initiatives that young learners have been actively engaged with over the past two decades. It is important to note that the projects subjected to analysis herein have already been documented in various publications. This study examines these publications to facilitate the desired analysis, thus enabling the creation of the envisaged visual model.

The projects subjected to analysis serve as distinct case studies, in accordance with Yin’s framework [40], which defines a case study as empirical research centered around a contemporary process within its real-world context, drawing from multiple sources of information. These projects were deliberately undertaken as case studies, given that their primary research questions fall under the scope of “Why?” and “How?” [40]. All of the case studies featured in this research study have been detailed in research papers or chapters in books, thereby facilitating the process of conducting a research synthesis.

A research synthesis encompasses a series of methods that serve to “summarize, integrate, combine, and compare the findings of different studies on a specific topic or research question” [41] (p. 441). The diverse case studies are subjected to scrutiny through a thematic synthesis process, facilitating the identification, analysis, and systematic organization of data-driven themes [42,43]. This thematic synthesis approach is particularly apt for assessing the significance of a specific subject of study (in this case, the use of sensors in environmental education projects) while also discerning the primary clusters of thematic elements, which can be visualized through a visual model [42].

Moreover, this research synthesis bears certain similarities to cross-case studies, as it employs visual aids such as tables and diagrams to analyze and visualize the shared themes and disparities across the various studies [43].

The methodology unfolds in several key steps. The outlining and description of the underlying dimensions of environmental education projects, as well as their influence in the use of sensors, is the first step, and this has already been presented in the previous section. In the second step, a preliminary exploratory analysis is undertaken to gain an overarching understanding of the dataset (the diverse projects) and to establish categories of thematic analysis based on the previously defined dimensions. The identification of these categories allows for an initial structured thematic analysis and the presentation of relevant themes within the context of the previously developed theoretical framework. Since the authors of this paper were involved in the projects that constitute the case studies, participant observation is a data collection method with foundational importance in this research study. The analysis of published papers, chapters, and sites where the projects were described constitute the *corpus* that enables bibliographic data collection. Tables are used in this research study both as analysis and visualization tools, and a bibliographic management tool (Zotero) was also used during our research.

The subsequent step involves scrutinizing the structured presentation to establish interconnections among the identified themes and categories, culminating in the inductive and iterative formulation of a visual model, using diagrams. This model’s development constitutes the final step of the process.

## 4. Results

This section presents the outcomes of our analysis of sensor use within a series of environmental education case studies. These case studies encompass projects that the authors of this paper have actively engaged with in Portugal over the past two decades.

Initially, all eight projects were examined based on the dimensions previously delineated in Section 2 (technological, social, didactic, and epistemic). Our aim was to pinpoint the relevant categories of analysis that could uncover both common and distinctive aspects in the use of sensors across each individual project or case study.

The selected categories for analysis encompass the following:Context/School grade: This category encompasses age, years of schooling, and their alignment with children’s cognitive development and problem-solving capabilities. The context, whether formal or non-formal, alongside the school grade, shapes the selection of strategies and resources (including sensors) and the techniques employed.Challenges to learners: These challenges are suggested practices aimed toward achieving learning outcomes such as acquired knowledge, attitudes, and competences.Main examples of developed competences: Competences represent a higher-level classification of learning outcomes. They are fostered through sensory and epistemic practices that leverage sensors within various activities, such as inquiries. Competences and practices combine to enable the creation of final products.Main examples of products: Products arise from sensory and epistemic practices and are developed to overcome the initial challenges set forth.Main used resources: This category encompasses a diverse array of resources, including electronic sensors, related apps, experiment plans, registration forms, data platforms, and other everyday and laboratory materials.

It is important to identify the close relations between the following dimensions and categories: (i) “technological dimension” and “main used resources”; (ii) “social dimension” and “context/school grade”; (iii) “didactic dimension” and “challenges to learners”; (iv) “epistemic dimension” and “main examples of products”.

Through the use of text and tables, as well as the above-mentioned categories, the following subsections present our analysis of the set of environmental education projects in consideration. The analysis is presented following the chronological order of the projects. The title of each of the following subsections epitomize the projects’ main characteristics, primarily with respect to the used hardware and software, the desired learning outcomes, and the didactic strategies of the projects.

### 4.1. Two Pioneering Projects

This section focuses on two pioneering projects, as presented in Table 1, which underscores the significance of precursor characteristics within the scope of the analyzed projects. The Globe Project is one of the more meaningful environmental education projects as it is known worldwide [44]. It is the only project presented in this paper in which the authors were not involved in. Originating in 1994 in the USA, this ongoing project engages schools worldwide in inquiry-based activities involving the collection and organization of environmental data [44,45]. The Globe platform facilitates the visualization of locally, regionally, and globally sourced data gathered by students [44]. The project’s learning outcomes encompass various disciplines, including natural sciences, mathematics, geography, and digital technologies, all stemming from inquiry activities within the local school environment. The Globe Project’s foundational components have left an indelible impact on subsequent environmental education initiatives that emphasize (i) challenging schools, and specifically students, to collect, treat, organize, and make sense of local environmental data (georeferenced in the context of the global environment through the use of the project’s data platform); (ii) fostering student agency and global citizenship participation; (iii) integrating both physical and virtual educational resources, including paper protocols, laboratory equipment, and a data platform.

The Senses@Watch project (Collaborative Environmental Monitoring: Tools and Models to Obtain and Analyze Environmental Information, POCTI/MGS/35651/1999) stands as a pioneer in harnessing environmental data acquired through human senses for citizen participation processes. This research project introduced novel methods for the concurrent application of human senses and sensors in the evaluation of environmental problems [46]. Additionally, the project developed a collaborative system to facilitate the creation of multisensory and multimedia georeferenced environmental complaints [46]. An educational approach to citizen participation was featured through the development of multimedia clipart designed to support the formulation of those complaints [47]. The outcomes of the Senses@Watch project left a noticeable imprint on the projects subsequently undertaken by the authors of this paper. This influence is particularly evident in the combination of sensory and sensor-based exploration and the assessment of the environment, as well as the creation of diverse representations of multisensory information.

During the 1990s and the early years of the 21st century, electronic sensors were not yet available for citizen participation and educational settings. Consequently, in the first decade of the Globe Project and within the Senses@Watch project, environmental data were acquired using non-electronic sensors such as colorimetric kits, thermometers, and other laboratory analysis tools [46].

### 4.2. Using Smartphones for Multisensory Exploration and Georeferenced Multimedia Communication: From Built-In to External Sensors

This section highlights two projects that employ smartphones for the multisensory exploration of school environments and georeferenced multimedia communication of these explorations (Table 2). The first project to be introduced, SchoolSenses@Internet, engaged primary school students in utilizing the built-in sensors of smartphones. Conversely, the second project, named USense2Learn, connected external sensors to an interface tailored for use by children, facilitating exploration and communication.

**Table 1 sensors-23-07677-t001:** Characterization of two pioneering projects in the use of sensors by children to characterize environmental systems.

Project	Context/School Grade	Challenges to Learners	Main Examples of Developed Competences	Main Examples of Products	Main Resources Used
The Globe Project (international project, started in 1994; https://www.globe.gov/, accessed on 26 July 2023).	All levels of education in formal and non-formal contexts.	− To follow protocols to collect environmental georeferenced qualitative and quantitative data; − To make sense of the collected data in local and global contexts through the visualization of the data collected by the other participants.	− Use sensors and other instruments to acquire quantitative environmental data; − Follow-up on the activities implemented by schools, which have changed over time: − Produce representations of sensors’ data; − Analyze collective data in diverse scales; − Develop an inquiry; − Relate environmental values.	− Georeferenced environmental quantitative and qualitative database; − Environmental inquires developed by schools; − Program’s environmental projects developed with schools; − Protocols; − Teachers’ guides; − Toolkit; − Interactive maps; − Globe platform.	− Sensors; − GPS; − Other and diverse laboratory analysis instruments; − Protocols; − Globe platform.
Senses@Watch (four-year project started in 2001) [46].	Population in general.	− To use senses to gather qualitative environmental information and assess and communicate environmental quality; − To use senses, along with sensors, to gather environmental information and assess environmental quality; − To create multisensory and multimedia messages based on environmental quality.	− Use senses to acquire and communicate qualitative environmental data; − Use senses to assess and communicate environmental data in multimedia messages; − Use senses, along with sensors, to gather environmental information and assess environmental quality.	− Multisensory qualitative environmental information; − Quantitative environmental information acquired by sensors; − Clipart to support citizens in creating multisensory and multimedia messages on environmental quality; − Multisensory and multimedia messages about environmental quality; − Assessment of the use of the senses to facilitate collaborative environmental monitoring.	− Colorimetric kits; − Official monitoring station of TRS concentrations; − Mobile phones; − Registration forms; − Laptops.

The SchoolSenses@Internet Project (Children as Multisensory Geographic Information Creators through the use of Information and Communication Technologies, POSI/EIA/56954/2004) explored the potential of georeferenced multisensory data to enhance learning within primary schools in the context of schoolyard environmental quality assessment [48]. This project developed a collaborative website accessible to primary schools that integrated Google Earth and a multimedia and multisensory message editor with clipart, also including modeling and simulation tools [49].

The advent of virtual globes, including Google Earth, in 2005 significantly impacted projects of this nature. While the proposal for the SchoolSenses@Internet Project preceded the emergence of Google Earth, the advent of virtual globes played a pivotal role in its execution. Its attributes, such as its ease of use, graphical fluidity, and rapid user feedback when creating virtual flights around the world, were instrumental. Additionally, Google Earth affordances to publish geographic, multisensory, multimedia environmental information in multiple representations proved crucial [48].

Prior to the era of virtual globes and ubiquitous mobile technologies, projects of geographic information creation and publication were documented. A notable historical precedent was the creation of the Domesday Book in the 11th century, a registry of English lands and properties initiated by William the Conqueror around 1086 (Domesday Book, n.d.). Nine-hundred years later, in 1986, the BBC introduced the Domesday disk, a laserdisc containing a survey of local geographic data in England carried out by children from thousands of schools, based on the division of the country into parts small enough to be inventoried locally with reduced human resources [50]. While the technology used in this BBC Domesday project—a voluntary geographic information project—became obsolete, its materials were made available in web format in 2011 [51].

In Portugal, the Internet@EB1 project launched in 2003 resulted in the creation and dissemination of nationwide voluntary geographic information provided by over 7000 primary schools [52]. This information focused on the local school community and environment, with georeferencing being based on official maps, children-designed maps, and orthophotomaps.

The SchoolSenses@Internet project was didactically centered around experiential learning strategies. Children used their senses and their smartphones’ built-in sensors, such as their phone’s GPS, microphone, and camera systems, to capture georeferenced multisensory environmental data while exploring their familiar schoolyards. Additionally, they crafted and sent multimedia messages that were compiled into KMZ files and subsequently published on Google Earth [53]. This publication not only facilitated data sharing but also facilitated a collective analysis of the messages to gain insights into environmental quality. In this project, multisensory environmental information was also used to enable children to use multisensory simulations and create multisensory environmental models [49]. Additionally, a multimedia editor equipped with a multisensory clipart prompted children to create multisensory narratives [54].

The USense2Learn project also centered on experiential learning strategies, inviting children to generate and send georeferenced multisensory multimedia messages to evaluate environmental quality using mobile phones. However, in this project, children were asked to explore and convey information on schoolyard thermal comfort [22]. To support this objective, the interface used for message composition included data obtained from a temperature and humidity sensor (external to the smartphone). A tailored platform was developed to facilitate these operations, involving a laptop, a smartphone for content creation, and at least one sensor, all equipped with GPS capabilities. The specific sensor used was the didactic (plug-and-play) PASCO PASPORT weather anemometer sensor (PS-2174).

**Table 2 sensors-23-07677-t002:** Characterization of the SchoolSenses@Internet and USense2Learn projects.

Project	Context/School Grade	Challenges to Learners	Main Examples of Developed Competences	Main Examples of Products	Main Resources Used
SchoolSenses@Internet—Three-year project started in 2005 [49,53,55].	Primary school children in outdoor curricular and extracurricular activities.	− Using your senses and a smartphone, describe your schoolyard; − Use clipart to produce multisensory and multimedia narratives and models; − Create a multisensory model of an environmental problem using *Simulkids* [55].	− Multisensory exploration and production of related multimedia recordings; − Use of a smartphone to take photos, record audio and video, and type text, as well as convey a message about the schoolyard; − Make sense of the diverse multisensory georeferenced messages, trough visualization in Google Earth; − To use iconic and symbolic elements to create multimedia narratives and models of environmental problems.	− Multisensory georeferenced messages with multimedia data from schoolyards; − Platform to produce a KMZ file that enables the visualization of children’s georeferenced multisensory messages in Google Earth; − KMZ file with the multisensory georeferenced messages to allow for visualization in Google Earth; − Clipart with multisensory elements; − Multisensory modeling tool for kids (*Simulkids* [55].), with models (entity relationship based) created by children.	− Arts and crafts to produce iconic and symbolic elements; − Smartphones with photo cameras, audio and video recorders, and GPS; − Laptops;
USense2Learn-1 year project, 2010. [22] (Silva, Lopes, Silva & Marcelino, 2010)	Fifth- and sixth-grade children in outdoor curricular activities.	− Using your senses, a smartphone, and temperature and humidity sensors to express the thermal comfort of diverse activities in your schoolyard.	− Multisensory exploration and production of related multimedia recordings; − Use of a smartphone to take photos, record audio and video, and type text while considering the temperature and humidity data to compose and send a message about the schoolyard thermal comfort; − Make sense of the diverse multisensory georeferenced messages, trough visualization in Google Earth.	− Multisensory georeferenced messages, with multimedia and sensor data from schoolyards; − Platform to integrate sensor data in the smartphone interface and produce a KML file that enables the visualization of the children’s georeferenced multisensory messages with sensor data in Google Earth; − KML file with the multisensory georeferenced messages and sensor data to for allow visualization in Google Earth;	− Smartphones with photo cameras, audio and video recorders, and GPS; − Plug-and-play sensors of temperature and humidity; − Laptops.

Both of the projects outlined in this section served as influential examples for subsequent projects. They illustrated the potential of utilizing smartphones’ built-in sensors to empower children in capturing georeferenced multisensory data. Furthermore, these projects illustrated the smartphone interface’s capability to compose and transmit georeferenced multisensory multimedia messages. The children successfully overcame the challenges presented by these projects, producing messages that represented the environmental quality of their respective schoolyards [22,48].

### 4.3. Using Senses and Didactic Sensors with Apps to Increase Participant Autonomy: From Primary School to University

This section presents two projects—SOS Abstract and Sensing Urban Places—that used didactic, affordable, and mobile electronic sensors in conjunction with tablet/smartphone apps. These projects challenge children and young individuals to engage with their environment by using their senses. The connection between sensory practices and epistemic processes is explicitly emphasized, with sensory data being linked to the quantitative information acquired by the sensors (Table 3). Both projects utilize smartphones with relevant apps and didactic external sensors as data loggers.

The first project, SOS Abstract (Using Sensors and Senses in the Environment to Develop Abstract Thinking), was tailored for primary school children aged 6 to 12 years old. The primary objective of SOS Abstract was to use both human senses and didactic sensors as epistemic mediators to cultivate abstract thinking [37]. Abstract thinking refers to the capacity to simultaneously hold multiple variables in mind or to comprehend the collective impact of input variables on a given outcome [56]. Human senses serve as the interface that directly engages with environmental variables and gradients, thereby forming a bridge between tangible sensory data and the more abstract data acquired by sensors related to environmental variables such as salinity, temperature, humidity, sound, light, and atmospheric contamination. SOS Abstract was a formal educational project that aimed to construct a connection between sensory perception and abstract thought among school children [37].

The SOS Abstract project employed PASCO PASPORT sensors (PASCO Scientific, Roseville, CA, USA) due to the fact that their capabilities aligned with the project’s objectives. These plug-and-play sensors are mobile, affordable, and robust. The specific sensors used included a weather anemometer sensor (PS-2174), a conductivity sensor (PS-3210), a PASPORT water temperature sensor (PS-2125), a PASPORT turbidity sensor (PS-2122), a PASPORT conductivity sensor (PS-2116), a pH sensor PASPORT (PS-2147), a digital microscope (USB 400X Microscope Veho® ROHS CE (Veho Southampton, UK) with 400× magnification, and the Samsung Galaxy S4’s sound sensor (Samsung Electronics, Suwon-Si, Gyeonggi-Do, Republic of Korea).

Didactically, the SOS Abstract Project employed various strategies across different activities, including experiential learning, research-based learning, and problem-based learning. It highlighted the interconnection between sensory and epistemic practices while generating diverse representations of the outcomes derived from such practices. One technique employed to establish this connection and foster abstract thinking is referred to as “concreteness fading”. This approach consists of successively decreasing the concreteness of the representations, to attain an abstract representation that is still connected to the situation represented [57]. Concreteness fading facilitates linking the visible to the invisible, the macro to the micro, or the qualitative (such as images and sound recordings) to the quantitative (data on environmental variables). The SOS Abstract project used concreteness fading in various scenarios, including the following: (i) linking macroscopic images to microscopic images using a digital microscope with continuous magnification, facilitating cognitive links between photos and drawings at various levels of magnification; (ii) connecting photos and textual descriptions of everyday scenarios and processes, such as combustion or crowded (non-)ventilated classrooms, with graphical representations of carbon dioxide concentration in the air (data acquired by sensors). In addition to foster abstract thinking, the project also nurtured epistemic practices and competences within the process of characterizing environmental systems. Notably, this focus extended to practices such as observation, data recording, and interpretation, all of which are crucial for facilitating abstract thinking development.

The second project—the Sensing Urban Places Project—involved aiming to enhance sensory engagement and convey sensations to architecture students through georeferenced multimedia messages complemented by sensor data. The students were prompted to utilize their senses along with (https://sensesandsensors.wordpress.com/, accessed on 26 July 2023) (i) in-built sensors on the Samsung Galaxy S4 smartphone (Galaxy Sensors app); (ii) the multimedia capabilities of the Samsung Galaxy S4, their Google account, and Picasa Web Albums; and (iii) a heart rate sensor (ZEPHYR Heart Rate Monitor HxM™ Bluetooth™, Veho, Southampton, UK) integrated with a smartphone through the Endomondo app to record tracks and heart rate data. The resulting georeferenced multimedia messages, tracks, and heart rate data were converted into kmz/gpx files, allowing for exploration and visualization within Google Earth.

An integral component of the Sensing Urban Places project was a workshop held at the Conference Linear Livability that centered around next-generation urban re-design multimodal green routes, hosted by Politecnico di Torino, Department of Architecture and Design. During this workshop, participating architecture students generated a series of georeferenced multisensory messages centered around Torino’s city center. These messages represented the students’ embodied explorations, encompassing visual, auditory, temperature, and humidity sensations, as well as heart rate data. Each sensory experience was linked to specific activities, such as crossing busy streets, biking, or relaxing near a fountain, all of which were contextualized within various urban routes. The production and analysis of these georeferenced multisensory messages facilitated the identification of urban challenges and potential solutions. This participatory workshop placed a significant emphasis on experiential and problem-solving learning.

These two projects, SOS Abstract and Sensing Urban Places, did not use ad hoc sites or platforms created for the projects. Instead, they made use of readily available and affordable everyday technology and free software. These projects exemplify how accessible and robust didactic sensors, combined with free apps functioning on ordinary mobile devices, can empower school children and young learners to autonomously foster learning outcomes and knowledge creation.

### 4.4. Using Senses and Didactic Sensors with Apps to Characterize Rural Environmental Cycles in Roleplaying Activities: The Dão Kids Academy Project

The Dão Kids Academy project was targeted at children aged 6 to 12 years old and aimed to encourage children to use both senses and sensors to explore the vineyards within the demarcated Dão region (Table 4). The project focused on characterizing the wine cycle while immersing participants in the roles of a farmer, reporter, and researcher [58]. Similar to the SOS Abstract and Sensing Urban Places initiatives, this project harnessed affordable and user-friendly mobile sensors (specifically, the PASCO PASPORT weather anemometer sensor PS-2174) coupled with a free app (SPARKvue) installed on mobile devices such as tablets and netbooks. This sensor and app combination facilitated the monitoring and understanding of meteorological parameters throughout the vineyards across the entire year [58]. Children were also equipped with action cameras and audio recorders to explore the environment and participate in farm-related activities, thus embodying their assigned roles [58]. Importantly, technology was seamlessly integrated into the context, functioning as professional tools to aid in the exploration of the environment and community, with the primary focus centering around the latter rather than the technology itself.

The project employed a research- and project-based approach to empower children to engage in epistemic practices and develop corresponding competences. Those practices included acquiring, organizing, and interpreting environmental data, as well as information revealed by farmers to children playing the roles of reporters and researchers. Children, in their roles as farmers, reporters, and researchers, used their senses while undertaking diverse activities within the vineyards, such as farming, conducting interviews, and discovering fresh perspectives through sensory engagement [58]. This immersive involvement within the environment and community served as a tangible foundation for understanding environmental data and understanding the relationships between variables and the wine cycle.

### 4.5. Using Didactic Sensors in Project- and Problem-Based Learning to Develop Environmental Citizen Participation: The Eco-Sensors4Health Project

The Eco-Sensors4Health project was established with the goal of assisting educators in guiding students to use their senses and employ sensors to identify, investigate, and address environmental problems within their school environment (Table 5). This approach aimed to foster participants’ sense of agency and environmental citizenship. The project placed a strong emphasis on cultivating project-based learning among students, facilitated through teacher mediation (https://eco-sensors4health.site/, accessed on 26 July 2023). To promote the integration of project-based learning, the proposed activities started with familiarizing students with prevalent school-related environmental problems, such as sound pollution, air pollution, and thermal discomfort, along with their underlying causes and impacts. Subsequently, children, in collaboration with their teachers, chose a specific problem area (such as sound pollution) and explored the phenomena in a multisensory way [59]. The process of identifying a specific problem, like sound pollution in a particular classroom, culminated only after a joint outlining of the experiment plan. This plan laid out how sensors such as sound sensors would be employed to establish independent variables and scrutinize their influence on dependent variables. Through data collection, recording, and interpretation facilitated by registration forms, environmental quality scales, and criteria, a comprehensive characterization of the targeted problem was achieved [59,60]. Proposed solutions were required to be both evidence-based and feasible, considering the resources of both students and teachers. The Eco-Sensors4Health toolkit was iteratively developed by the project’s research team to provide comprehensive support for teachers and students engaging in the described project-based, inquiry-based, and problem-based learning [61].

The developed Eco-Sensors4Health platform includes a data input section where students and teachers can enter the environmental health data they acquired using sensors. Furthermore, it includes a query area that enables the retrieval of previously submitted data (collected by students). This platform also facilitates the submission and retrieval of inquiry documents such as experiment plans, photographs, and textual descriptions. The design process for this platform was iterative, including characteristics similar to that of design-based research, and led by a multidisciplinary team with expertise in both ICT and education [62].

Within the Eco-Sensors4Health project, the following PASCO PASPORT sensors were used by students: the weather anemometer sensor (PS-2174), the sound level sensor (PS-2109), the carbon dioxide sensor (PS-3208), and the light sensor (PS-2106A). As previously noted, these sensors were intentionally chosen due to their ease of use, mobility, affordability, and robustness. Students were able to independently operate these sensors, while the free SPARKVue app, installed on tablets, functioned as an efficient data logger, allowing students to monitor and visualize data even while on the move.

The activities outlined within the Eco-Sensors4Health project were initially designed for primary school students. However, their successful implementation extended beyond this scope and reached high school and higher education levels, including teacher training [63]. This project contributed to the identification, exploration, and solving of environmental problems pertaining to the school through participants’ agency and environmental citizenship [60].

**Table 3 sensors-23-07677-t003:** Characterization of the SOSAbstract and Sensing Urban Places projects.

Project	Context/School Grade	Challenges to Learners	Main Examples Of Developed Competences	Main Examples of Products	Main Resources Used
SOSAbstract (four-year project started in 2010) [37,64].	Children from 6 to 12 years old. Curricular activities indoors and outdoors.	Using senses and sensors, characterize an environmental system, producing recordings and representations of the multisensory and quantitative observations.	− Develop multisensory environmental exploration; − Produce representations of environmental multisensory information; − Use of plug-and-play sensors to acquire quantitative environmental data; − Produce representations of sensor data; − Annotate representations of sensor data with multisensory information; − Relate multisensory information and sensor data to environmental characteristics, processes, and quality; − Relate environmental variables.	− Representations of multisensory information of situated environmental systems; − Registration forms to support learners in epistemic practices, such as observation and data registration and interpretation; − Quantitative data of situated environmental systems; − Representations of relations between sensor data and multisensory environmental information; − Concreteness fading representations of environmental characteristics.	− Plug-and-play sensors of diverse environmental variables, such as temperature, humidity, CO_2_ in air, conductivity, pH, turbidity, light, and sound; − Microscopes; binocular loupes; − Laptops.
Sensing Urban Places (one-year project, 2015). (https://sensesandsensors.wordpress.com/, accessed on 26 July 2023).	Higher education.	Describe sensing experiences in the city, through the production of georeferenced multisensory information enhanced with information acquired by electronic sensors. Your work will be disseminated as georeferenced multimedia messages.	− Develop multisensory environmental exploration; − Produce representations of georeferenced environmental multisensory information enhanced with sensor information; − Use a smartphone to take photos, record audio and video, and type text to compose and send a message about sensing experiences in the city; − Make sense of the produced messages through visualization in Google Earth.	− Representations of georeferenced environmental multisensory information enhanced with sensor data; − Qualitative and quantitative data of situated urban environments; − Representations of relations between sensor data and multisensory urban experiences; − Maps with messages presenting georeferenced environmental multisensory information enhanced with sensor data.	− Smartphones with integrated (via apps) sensors of diverse environmental variables, such as sound, temperature, humidity, and heart rate; − Laptops.

**Table 4 sensors-23-07677-t004:** Characterization of the Dão Kids Academy project.

Project	Context/School Grade	Challenges to Learners	Main Examples of Developed Competences	Main Examples of Products	Main Resources Used
Academia Dão Petiz (Dão Kids Academy)—one-year project started in 2015 [58].	Primary school children and fifth- and sixth-grade children in non-formal outdoors activities.	− Explore agricultural cycles representative of the students’ geographic region (vineyard, linen); − Play the role of a farmer; − Play the role of a researcher; − Play the role of a reporter.	− Experience the work of farmers, hands on activities in the vineyard in the following representative stages: pruning, binding of the shoots, harvest, post-harvest; − Characterize environmental indicators throughout the wine cycle using senses and sensors and discern their relation to the vegetable cycle; − Describe the vine cycle by organizing short reports about the activities in the farms.	− Representations of multisensory information along the agricultural cycle; − Qualitative and quantitative data regarding the farm environment along the agricultural and transformation cycle; − Short reports with images and interviews with the farmers and other workers, as well as other children playing the role of farmers or researchers.	− Arts and crafts to produce notes, drawings, storyboards to produce reports and design table sheets to collect data; − Plug-and-play sensors of diverse environmental variables, such as temperature, humidity, and wind speed. − Action cameras and audio recorders.

**Table 5 sensors-23-07677-t005:** Characterization of the Eco-Sensors4Health project.

Project	Context/School Grade	Challenges to Learners	Main Examples of Developed Competences	Main Examples of Products	Main Resources Used
Eco-Sensors4Health (three-year project started in 2017) [59,60,63].	All school grades. Curricular activities indoors and outdoors.	Explore a local school environment using senses and sensors to identify and solve a problem using an experimental inquiry method.	− Make an experiment plan to identify and solve an environmental problem that pertains to the school using senses and sensors; − Develop a multisensory exploration of the school’s environmental problem and communicate the results (sensory practices); − Implement the experimental plan (use of plug-and-play sensors) to acquire quantitative environmental data and register, classify, and interpret the data; − Based on the acquired data, make a decision to solve the school’s environmental problem; − Communicate the decision; − Implement the decisions made.	− Experiment plans to identify and solve an environmental problem pertaining to the school and using senses and sensors; − Qualitative data of the school’s environmental problems; − Quantitative data of the school’s environmental problems; − A set of real school environmental problems; − A set of solutions for real school environmental problems; − Solved School environmental problems. − Eco-Sensors4Health collaborative data platform; − Eco-Sensors4Health Toolkit.	− Plug-and-play sensors of diverse environmental variables, such as temperature, humidity, CO_2_ in air, light, and sound; − Environmental quality scales and criteria; − Laptops.

## 5. Discussion

In addition to the presentation of two pioneering environmental projects that illustrated the potential of senses and sensors, the previous section introduced six projects that were developed with the active involvement of the authors of this paper. These six projects share several common characteristics, including the following:The integration of Senses and Sensors: All of the projects involved participants, including children and youngsters, in sensory practices and the use of electronic sensors for epistemic practices.Learning Outcomes: A primary goal across all projects was the attainment of specific learning outcomes, including acquired knowledge, attitudes, and competences aligned with environmental education objectives.Teacher Mediation: All project activities, including sensory and epistemic practices, were conducted under the mediation of teachers, ensuring the proper facilitation of active participant engagement and empowerment.Participants’ agency in the sensory and epistemic practices promoting empowerment.Use of Mobile Devices and Technology: Mobile devices such as sensors, smartphones, and tablets were integral to environmental exploration and communication, aiding both sensory and epistemic practices. Each project involved the creation and use of supportive resources such as registration forms and everyday objects to enhance the exploration of physical environmental phenomena such as sound.Creation of Situated Knowledge: The projects contributed to the generation of context-specific knowledge and geographic information.

Despite these shared characteristics, it is important to scrutinize the six projects for distinctive features, considering the dimensions of environmental projects as discussed in Section 2. In terms of environmental education goals, all projects sought to enhance participants’ environmental knowledge. In addition, the SchoolSenses@Internet, USense2Learn, Dão Kids Academy, and Sensing Urban Places projects aimed to cultivate environmental awareness and qualitative and quantitative characterization and communication competences concerning local environments.

Similarly, the SOS Abstract project, alongside its knowledge acquisition objectives, placed emphasis on fostering competences that linked concrete practices to abstract environmental information and representations. Additionally, the Eco-Sensors4Health project aimed to address every category of the environmental education goals, from awareness, through competences of environmental problem-solving, to participation in environmental improvement and protection.

From a technological perspective, the projects followed the evolution of technology, and this is reflected in their utilization of mobile sensors, apps, mobile phones, tablets, netbooks, and communication platforms. While the SchoolSenses@Internet and USense2Learn projects employed prototype platforms that intertwined various functionalities, the Sensing Urban Places project effectively leveraged existing free apps and services, thereby obviating the need for technical support during task execution.

The Dão Kids Academy and SOS Abstract projects used mobile sensors that were linked to an app (functioning as a data logger) installed on a smartphone or tablet, enhancing robustness, affordability, and ease of use for children, youngsters, and teachers. In these projects, the technology used was designed to be transparent, with innovation primarily concentrated on educational and sensor application aspects.

The Eco-Sensors4Health project used the same mobile devices as the Dão Kids Academy and SOS Abstract projects but introduced an ad hoc collaborative data-sharing platform. This collaborative platform was iteratively developed by a multidisciplinary research team, enabling children to input, query, and visualize data concerning the environmental problems addressed by different participating schools [62]. This platform was implemented and used in the project activities with children and teachers [60].

Chronologically, the first two of the six projects in consideration were created to induce innovations in technology to enable the production and processing of georeferenced multisensory multimedia messages. Technological advancements allowed the subsequent projects to focus solely on educational innovation. The creation of the Eco-Sensors4Health platform exemplifies the application of pre-existing technology to achieve educational objectives. Notably, the latter three projects operated without prototypes, enabling autonomous and easily replicable technology use by both children and teachers.

From a didactic standpoint, the earlier projects adopted an environmental exploratory approach, while the more recent ones embraced a participatory methodology, mirroring contemporary societal trends that emphasize environmental citizenship and local-global participation. In the SchoolSenses@Internet and in the USense2Learn projects, the focus was on experiential learning. However, children’s actions were not completely autonomous, and they needed more real-time support and guidance. In contrast, the Dão Kids Academy, SOS Abstract, and Sensing Urban Places projects were based on technology robustness and ease of use, enabling increased autonomy among children and affording teachers the opportunity to center their mediation on environmental education objectives. These three projects employed a combination of experiential learning, research-based learning, and problem-based learning strategies. The Eco-Sensors4Health project, epitomizing participatory engagement, granted participants greater autonomy, allowing for teacher’s to adopt a more flexible attitude, with project and problem-based learning emerging as privileged strategies. Consequently, this recent project targeted higher-level environmental education goals, integrating strategies that demanded greater autonomy and complex actions, thereby fostering diverse learning outcomes and greater knowledge creation.

Across the spectrum of the six analyzed projects, there is a discernible progression from multisensory observation and communication (the SchoolSenses@Internet and USense2Learn projects) to data collection, representation, and interpretation (the Dão Kids Academy, SOS Abstract, and Sensing Urban Places projects), culminating in the planning and implementation of experimental inquiries, variable control, and decision-making processes (Eco-Sensors4Health project).

## 6. The SEAM Model

The analysis and subsequent discussion of the research synthesis findings culminated in the identification of particular themes and subjects which were subsequently structured into a visual model named SEAM—Sensors and Senses in Environmental Education Along with Teacher Mediation (Figure 1). This model was developed to inform the design of environmental education projects in which children and youngsters use senses and sensors as epistemic mediators.

In this model, the use of sensors is part of the learners’ environmental educational practices. Sensors can be used as extended senses in sensory practices and as tools to learn and create knowledge in epistemic practices [38]. Both of these kinds of learners’ practices can produce learning outcomes and create knowledge.

Learning outcomes can include the acquisition of knowledge, as well as the development of attitudes and competences. Moreover, the knowledge generated can exhibit diverse levels of theoretical abstraction, with situated knowledge occupying a more empirical stance. Situated knowledge can be systematically organized and mapped, constituting more abstract information such as Voluntary Geographic Information, and theorized in case studies.

To strategically gear learners’ practices towards the attainment of specific learning outcomes and diverse types of knowledge creation, it becomes essential to define the categories of the environmental education goals [65] that are being pursued. Environmental education goals span a spectrum. Awareness of and sensitivity to the environment and its problems are basic goals in environmental education, while skills and competences are higher-level goals. Participation, on the other hand, is in the highest category of the environmental education goals, according to the Tbilisi Conference [65]. The process of developing awareness, skills, competences, and participation invariably involves the acquisition of knowledge. As one progresses from lower-level goals like knowledge acquisition to higher-level objectives like participation, an augmented degree of learners’ autonomy becomes imperative.

Didactic strategies combine methods, processes, techniques, and organization approaches, including the use of sensors, to make it possible to achieve the defined goals [66]. The distinct categories of goals outlined in the model correspond to specific didactic strategies, effectively delineating both teacher mediation and learners’ agency in the promotion of designed learning practices using sensors [66]. For instance, experiential learning is aptly suited to exploratory learners’ practices, while the pursuit of participation logically aligns with project-based or problem-based learning methodologies. The complexity of foreseen teacher mediation and learners’ practices progressively increases through the array of didactic strategies presented within this model, showing a continuum from experiential learning through research-based and problem-based learning and culminating in project-based learning.

Teacher mediation transforms sensors and complementary support resources into epistemic mediators, guiding and helping learners through sensory and epistemic practices. This instructional approach serves to bridge learning gaps and surmount challenges such as the continuous integration of tangible experiences with abstract concepts and information. Teacher mediation and learners’ agency unfold in an interactive dynamic before (in the specification of goals, strategies, and in the planning stages), during, and after learners’ practices (during final assessment and communication stages).

In addition to senses and sensors, the inclusion of other support resources is due to their role in teacher mediation and enhancing the agency of learners in sensory practices and, especially, in the realm of epistemic practices. Notably, everyday household and laboratory items, as well as tools like data platforms and mobile ICT devices, play a vital role. Furthermore, an assortment of documents, ranging from registration and interpretation forms to multimedia presentations and various other multimedia creations, further contribute to the efficacy of teacher mediation and learners’ autonomous engagement in the learning process.

## 7. Concluding Remarks

The research presented in this paper centers around the use of electronic mobile sensors by children and youngsters in environmental education projects in Portugal. The analysis of the use and of the role of the sensors in a set of environmental education projects included a research synthesis of such projects, which were investigated as case studies.

Through examining these environmental education projects in chronological order, we observed that the following overarching trends emerged: (i) The progression towards greater sensor robustness, user-friendliness, complexity, and specificity. (ii) The rise in learners’ autonomy, both in terms of defined goals and executed practices involving senses and sensors. (iii) The increasing grade of complexity in the educational strategies. (iv) The increasing level of abstraction in learning outcomes and in the created knowledge.

Our analysis of the goals, strategies, and practices of learners with respect to the environmental education projects discussed in this paper revealed their relation to the environmental education currents defined by Sauvé [67].

All the projects, starting with the SchoolSenses@Internet and the USense2Learn projects, strengthened the importance of practices such as the use of sensors in the environment, thus showing a close relation to the experiential dimension of the naturalist current;The SOS Abstract and the Eco-Sensors4Health projects used sensors in problem-based learning strategies, following the foundational problem-solving current of environmental education;In the Dão Kids Academy project, the use of sensors by children followed the bioregionalist current of environmental education since the project prioritized relationships to the environment and the community, developing a sense of belonging;In the Sensing Urban Places project, architecture students used sensors to enable autonomous, sensory, epistemic, creative, and effective environmental exploration, as preconized by the holistic current;In the context of problem- and project-based learning, the use of sensors in the Eco-Sensors4Health project was based upon action and the improvement of action in the school environment, developing the main goals of the *praxic* current.

Throughout these projects, sensor utilization varied in a set of dimensions, as elaborated in this paper. It is worth noting that cases where sensors were innovatively integrated to create novel functionalities and communication pathways, often through prototypes, required a trade-off between sensor robustness and child autonomy, favoring technological innovation over lower-level educational innovation. Conversely, when the projects utilized commercialized systems, the children’s autonomy enabled more profound and deep-rooted educational innovation.

The SEAM model (Sensors and senses in Environmental education Along with teacher Mediation) was developed based on the research presented in this paper (a research synthesis that involved characterizing selected case studies). This model is designed to support environmental educators in the design and implementation of environmental education projects where the use of the human senses and sensors plays a central role. Environmental projects that adhere to the SEAM model may exhibit varying characteristics in terms of their goals, strategies, learners’ practices, resources, learning outcomes, and knowledge creation.

## Figures and Tables

**Figure 1 sensors-23-07677-f001:**
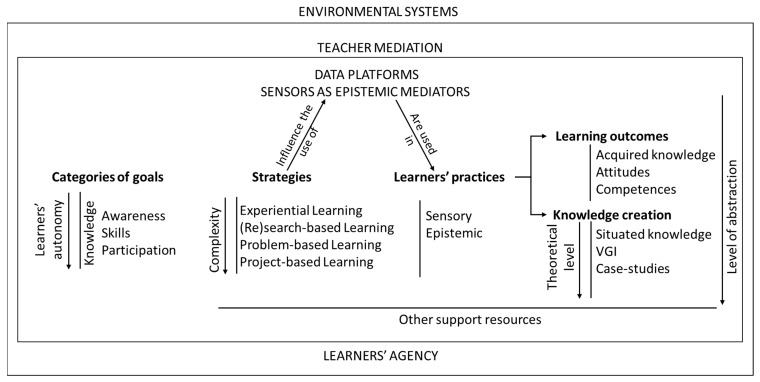
The SEAM model (Sensors and senses in Environmental education Along with teacher Mediation).

## Data Availability

The investigation described in this article is a research synthesis of projects/cases previously developed and published. No new data were created in this study. Data sharing is not applicable to this article.

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
