# Peer review of "The Use of Mobile Sensors by Children: A Review of Two Decades of Environmental Education Projects"

_sensors, 2023, doi:10.3390/s23187677_

Round 1
Reviewer 1 Report
The authors should present more clearly the research objectives, design, data collection tools and procedure.
The results have tables that could be summarized and explained coherently in the text. The tables can be added as an appendix.
Although the line of research and the work done on the case studies are coherent, it would be necessary to include a line of argumentation that would facilitate understanding based on the scope of use of sensors.
The writing is correct and comprehensive.
Author Response
Thank you very much for your comments and suggestions.
1 - The text has been revised to improve the quality of English in this paper.
2 - "The authors should present more clearly the research objectives, design, data collection tools and procedure." - the Research design section was improved to meet these requirements.
3- "The results have tables that could be summarized and explained coherently in the text. The tables can be added as an appendix." - the introductory part of the "Results" section was improved to facilitate the reading of the mentioned section.
4 - "Although the line of research and the work done on the case studies are coherent, it would be necessary to include a line of argumentation that would facilitate understanding based on the scope of use of sensors." - the changes in the sections "Research design" and "Results" (introductory part) were made to facilitate understanding.
Reviewer 2 Report
Dear authors,
Congratulations for such a paper. I think it is well written and referenced. It includes an overview of several educational projects related with the use of sensors for environmental education. Moreover the final SEAM model that you propose can be seen as a good tool for teachers when developing educational activities and projects in this subject
Author Response
Thank you very much for your comments.
Reviewer 3 Report
the article should be improving. the main idea is difficult to find and the source of projects is losing in too many details.
difficult to understand
Author Response
Thank you very much for your comments and suggestions.
1 - The text has been revised to improve the quality of English in this paper.
2 - "the article should be improving. the main idea is difficult to find and the source of projects is losing in too many details" - the Introduction, the Research design section, and the introductory part of the "Results" section were improved to make the paper easier to read.
Reviewer 4 Report
Authors have done quite some strenuous work here and the mode of address on the subject is commendable.
- the objectives and the rationale of the study are clearly stated.
however, it will be good if they clarify the significant themes and flow to enable fresh readers to move along with the research as it builds upon previous outcomes from this same investigation.
for studies such as this that are very contemporary and cut across different continents in their usefulness, it could be suggested the authors do some theoretical and empirical layouts to back up the essentiality of this research
though the analysis and discussion of the results of the developed research synthesis allowed the identification of themes and topics, and their organization in a visual model, having the clarification of a brief in theoretical, and empirical backings would do a better understanding for readers.
- The research design was adequate and appropriate for such studies. the research synthesis approach used by the researchers is very engaging as it spans through the use of different methods and at the same time summarizes, integrates, and combines the outcome of different stages is robust and as well elaborates the results as regards the questions of the research as displayed. this is commendable.
- the results of the analysis of the use of sensors in a set of environmental
education case studies for the past 20 years as presented are clearly indicative of the rigorous steps undertaken by the researchers.
the results are well presented, and the dimensions considered were adequately outlined.
the discussions are well outlined and elaborately explained in line with the previous projects in a chronological manner and the visual expressions alongside the figures.
the communication is highly technical and easy to understand.
-authors are encouraged to do a little editing check on the manuscript before final publication
Author Response
Thank you very much for your comments and suggestions.
1 - The text has been revised to improve the quality of English in this paper.
2 - "however, it will be good if they clarify the significant themes and flow to enable fresh readers to move along with the research as it builds upon previous outcomes from this same investigation.
for studies such as this that are very contemporary and cut across different continents in their usefulness, it could be suggested the authors do some theoretical and empirical layouts to back up the essentiality of this research
though the analysis and discussion of the results of the developed research synthesis allowed the identification of themes and topics, and their organization in a visual model, having the clarification of a brief in theoretical, and empirical backings would do a better understanding for readers" - the Introduction, the Research design section, and the introductory part of the "Results" section were improved to make the paper easier to read by "fresh readers" and enhance understanding.
Round 2
Reviewer 1 Report
The manuscript has been sufficiently improved to warrant publication in Sensors.
The manuscript has been sufficiently improved to warrant publication in Sensors.